# Topological radiation from vortex masers

Haoye Qin [1,2], Rongrong Xiang [1,2], Amir Jafargholi[1], Zhe Zhang [1] &
Romain Fleury [1] ✉

Vortex singularities in acoustic and electromagnetic fields are instrumental
degrees of freedom in advanced wavefront-shaping schemes and robust high-
throughput communications. Laser sources that emit coherent vortices in
free space have been demonstrated at optical frequencies, however their
microwave counterparts, *vortex masers*, have remained entirely unexplored,
despite their promising application potential as low-noise quantum sources
and sensors. Here, we demonstrate a room-temperature maser emitting pulses
of electromagnetic radiation with polarization and phase vortices, based on
the physics of 3D topological vectorial singularities. Nontrivial microwave
photons with polarization winding are emitted from a maser made of a sub-
wavelength dielectric cavity filled with an organic gain medium. By topping the
cavity with a chiral metasurface, the circular polarizations decouple, allowing
the masing of pulses with nonzero orbital angular momentum revealed
through nontrivial wavefront winding. Our work paves the way for multi-
dimensional vortex and singularity emission from volumetric coherent
microwave sources, topological photonic radiation, and novel practical
applications of masers.

Vortices are topological singularities in complex fields that sig-
nificantly expand the spatial degrees of freedom of waves, enabling
transformative applications in high-capacity communications, super-
resolution imaging, and particle trapping[1–4]. Vortex generation has
been demonstrated across diverse planar architectures, including
spatially distributed metasurfaces[3,5–7], photonic crystals[8–10], bound
states in the continuum[11–15], and ring resonators[16–19], spanning from
conventional acoustic setups to cutting-edge nano-optical and atomic
platforms[20].

Typically, these implementations feature large 2D planar surfaces
that generate vortices in the normal direction based on the dispersion
of collective modes or large-scale resonances. On the other hand,
vortex emission from a subwavelength emitter presents greater chal-
lenges due to the small volumes involved, and the required extension
of coherent vortex generation mechanisms beyond 2D[8,21–23]. Such
dimensional extension of topological singularities and of their appli-
cations is in general nontrivial, and is at the heart of current research in
topological physics[24–26].

Regarding wave-emitting devices, vortex generation has been
primarily focused on the optical regime through photoluminescence

and lasing mechanisms. However, masers[27–29], the microwave pro-
genitor of optical lasers, despite their broad applications in fields such
as deep-space communications[30], astronomy[31], and quantum
sensing[32,33], have so far been restricted to the emission of topologically
trivial fields, and have not yet been empowered by more advanced
functionalities such as vortex emission in one or several simultaneous
directions.

Here, we demonstrate a pulsed multi-directional vortex maser
based on a cubic subwavelength dielectric cavity filled with pentacene-
doped p-terphenyl (pc:pt) crystal as the gain medium[27,34,35], which
exhibits emission of microwave photons with topological polarization
vortices perpendicular to each of its faces. Direct observation of pulses
with orbital angular momentum (OAM) is realized by coupling this
cavity with a spin-selective metasurface. Additionally, we verified the
simultaneous coexistence of masing signals in three orthogonal
directions and the associated singularities induced by the vortices.
This extends the principles of active topological photonics, previously
confined to optical lasers, into the microwave regime, where coherent
emission of this complexity has not been achieved. It provides critical
insights into how such coherent states can enhance performance,

[1]Laboratory of Wave Engineering, École Polytechnique Fédérale de Lausanne, Lausanne, Switzerland. [2]These authors contributed equally: Haoye Qin,
Rongrong Xiang. ✉e-mail: romain.fleury@epfl.ch

reduce noise, and enable more precise control over electromagnetic fields.

We believe that such multi-directional vortex masers can provide new degrees of freedom in a next generation of volumetric coherent microwave sources[36], by exploiting multidimensional topological vortices and singularity emission in quantum sensing or communication applications.

## Results

### Concept of topological multi-directional vortex maser

Figure 1a presents the topological multi-directional vortex maser, constructed around a symmetric, standalone cubic strontium titanate (STO, with relative permittivity $\varepsilon_r = 318$) cavity of deep-subwavelength size, within which cylindrical monocrystals of pentacene-doped p-terphenyl are inserted, serving as the gain medium necessary for masing at room temperature. Upon pumping with yellow light (592 nm), population inversion can be established and stimulated emission can occur[29,34], leading to the radiation of microwave vortex photons from each surface of the cube in the normal direction. This is enabled by the special eigenmode of the symmetric cube, which we designed to support topological polarization singularities along these six directions (see Supplementary Fig. S1).

The cubic STO cavity supports a unique eigenmode in which the electric field intensity vanishes at the center ($E_x = 0$ and $E_y = 0$ forming nodal lines and crossing at the center point as vectorial zero), while the polarization angle $\phi_p = \arg\left(E_x + iE_y\right)$ undergoes a continuous $2\pi$ winding on planes parallel to the cube faces. The drilled voids concentrate magnetic fields, ensuring efficient coupling with the spin-polarized triplet transitions in the gain medium. This synergy between cavity symmetry, vectorial topology, and gain–cavity interaction constitutes the physical foundation of the maser's multi-directional vortex emission. This provides a physical picture of the mechanism underlying the 3D topological singularities, underscoring the novelty of our work: the first realization of a coherent volumetric source where topological polarization singularities are intrinsically embedded in the eigenmode itself, rather than imposed externally.

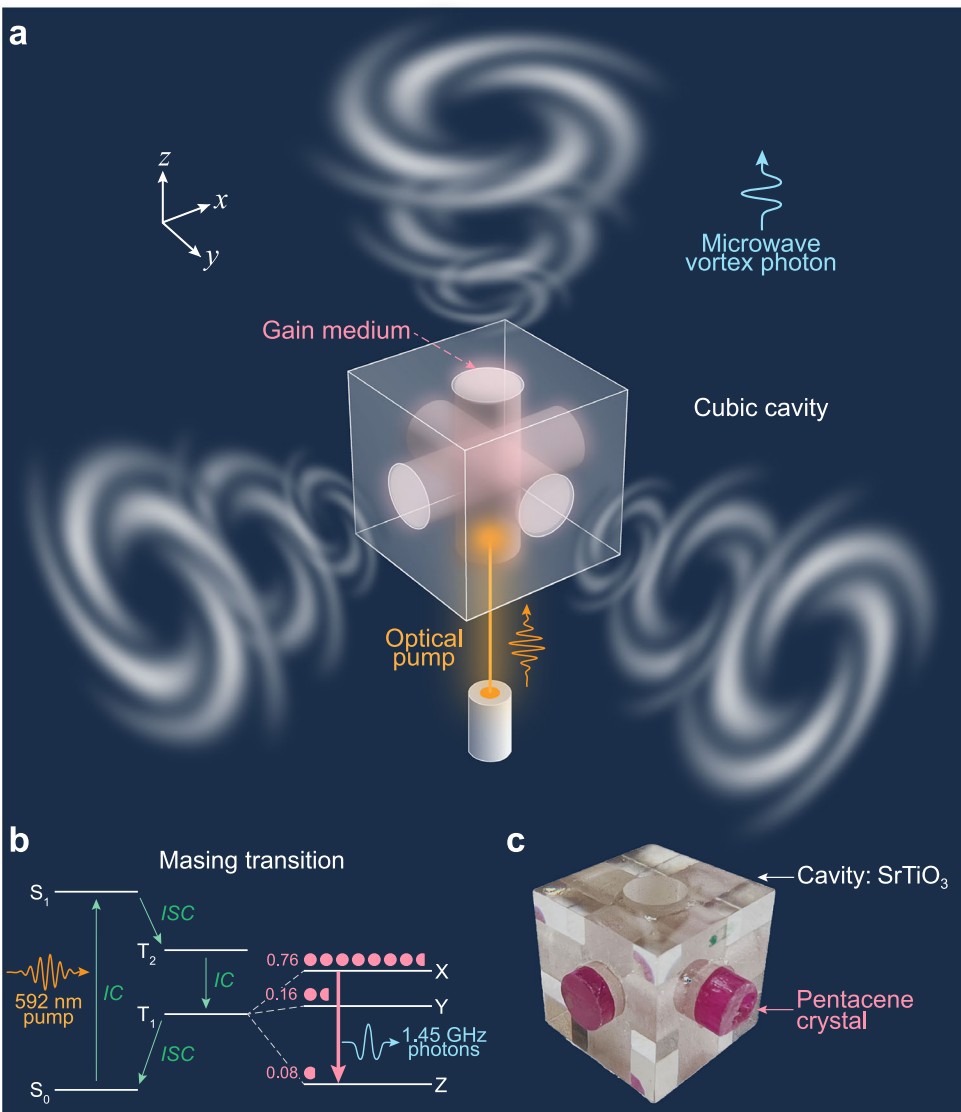

**Fig. 1 | The topological multi-directional vortex maser. a** Illustration of topological vortex masing actions toward every direction normal to the cubic resonator. The cubic maser containing the gain medium is subjected to an optical pump from an optical fiber positioned beneath the system. Microwave vortex photons are emitting coherently in multiple directions. **b** Masing transition scheme: The Jablonski diagram illustrates the masing process in pentacene-doped p-terphenyl, which functions as the gain medium. Population inversion between the X and Z energy levels enables the masing transition, resulting in the emission of coherent microwave photons at approximately 1.45 GHz. **c** Photograph of the fabricated subwavelength cubic cavity made of dielectric strontium titanate featuring three through holes. The cylindrical pentacene crystal as gain medium is inserted into each air hole, forming a compact integrated device.

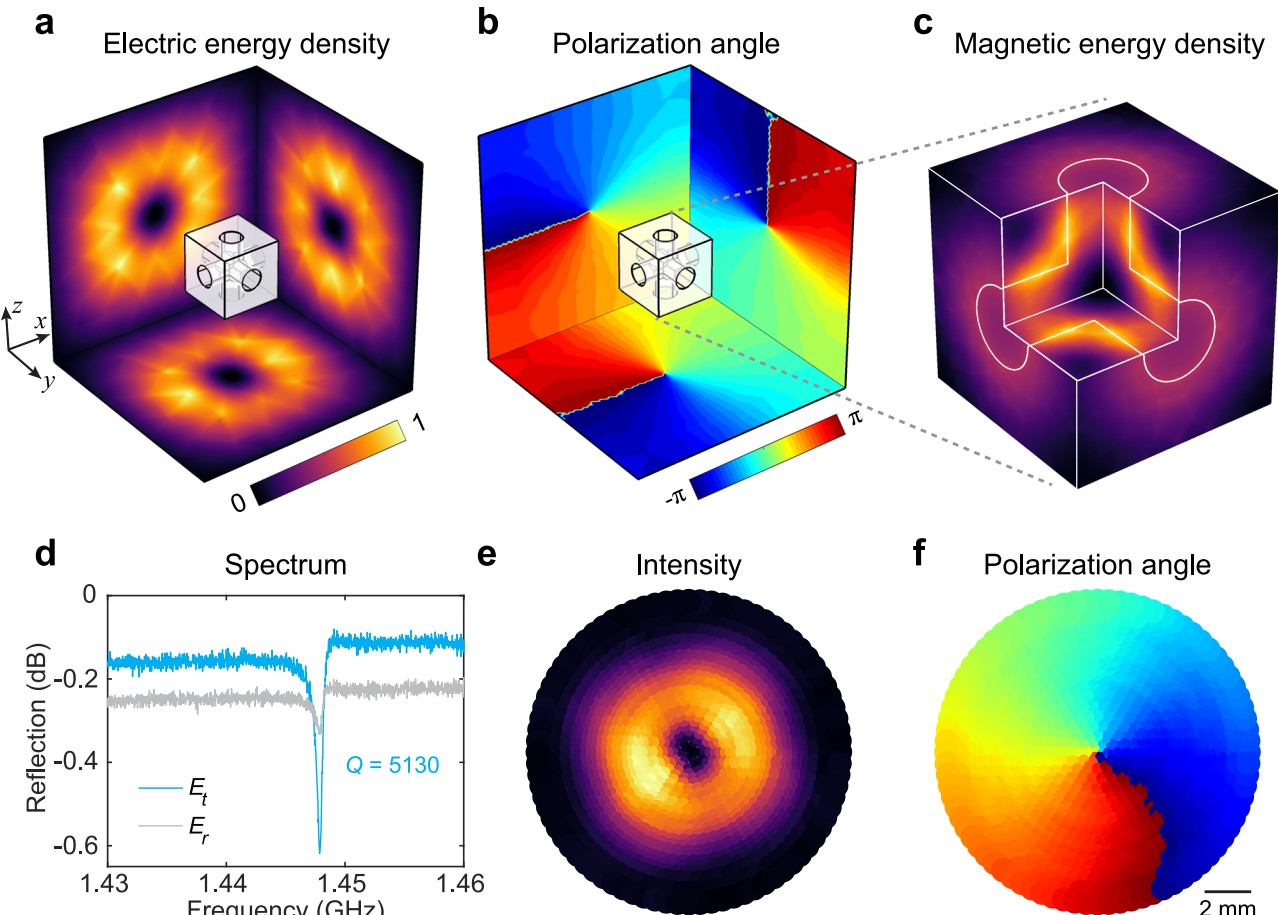

**Fig. 2 | Topological polarization singularities. a, b** Simulated electric energy density profile (**a**) and polarization angle (**b**) of the maser eigenmode, with cut planes in three directions. The cavity has a side length of 14.1 mm and a hole radius of 2.5 mm. **c** Corresponding magnetic energy density, confined within the holes filled with the gain medium. **d** Experimentally measured reflection spectrum of two orthogonal electric probes at the tangential ($E_t$) and radial direction ($E_r$), revealing the presence of the maser mode with a $Q$ factor of 5130. The use of two probes allows the extraction of the direction of polarization state. **e, f** Experimentally measured electric field intensity (**e**) and corresponding polarization angle (**f**) distribution on a plane parallel to one of the cube's surfaces. A non-trivial polarization winding of charge 1 is observed around the intensity singularity.

The transitions leading to masing within the gain medium are known and well documented in other works[27,37–39], however, for the reader's convenience, we reproduce a simplified Jablonski diagram in Fig. 1b. The 592-nm optical pump initially excites pentacene molecules from the ground state $S_0$ to the excited state $S_1$ through internal conversion (IC). About 62.5% of these molecules cross to triplet state $T_2$ with spin polarization (known as intersystem crossing), then relax to sublevels X, Y, and Z within state $T_1$. This creates a population inversion with ratios of 0.76:0.16:0.08. When the gain medium is placed within a microwave cavity resonant at approximately 1.45 GHz, stimulated maser transitions occur between the X and Z sublevels, depleting the population inversion and generating coherent microwave emission.

Figure 1c displays a photograph of the fabricated multi-directional vortex maser, featuring a STO cubic cavity with three identical cylindrical holes drilled perpendicular to each pair of opposing faces. Cylindrical pieces of the gain medium are loaded into these holes, forming a compact integrated device with dimensions of ~0.07 wavelengths. Through an intuitive and accessible design, this symmetrically configured system achieves enhanced volume-level light-matter interaction for topological wave emitting, as we show below.

**Eigenmode with topological singularities**
Figure 2 characterizes the cubic cavity eigenmode that exhibits multiple topological polarization singularities arising from the localized resonance (see also Supplementary Fig. S1 for additional details). It's noteworthy that the functionality of the air holes is twofold: they create a special resonance that hosts polarization singularities along all three orthogonal directions (normal to the cube faces), while simultaneously providing a void space with concentrated magnetic field, maximizing the interactions with the magnetic dipoles in the gain medium. The topological singularities in our system are intrinsic to the eigenmode itself, providing enhanced flexibility and stability compared to systems where the topology is externally imposed.

When viewed at any cut plane parallel to the cube's surface, the simulated electric field profile consistently exhibits a ring-shaped distribution with a zero singularity at the center and high amplitude in the surrounding region (Fig. 2a). The resultant polarization angle map reveals a 2π winding around each singularity (corresponding to a topological charge of ±1), confirming the vortex nature of the electric eigenmode in multiple directions (Fig. 2b). These zero-intensity singularities are real-space topological defects of the electric vector field immersed in 3D space: the total instantaneous polarization vorticity over any surface enclosing the resonator is 2, consistent with the Poincaré-Hopf theorem. Additionally, the corresponding magnetic field is strongly enhanced and confined within the air holes (Fig. 2c; see Supplementary Fig. S4 for cut plane view of field localization), facilitating the interaction between cavity resonance and gain medium for masing action throughout the device volume.

To validate these results, we experimentally measured the reflection spectrum using two orthogonal electric probes (Fig. 2d). The

probe orientation is shown in Supplementary Fig. S2a. Due to the strong spatial polarization anisotropy of the eigenmode, the tangential and radial probes will exhibit markedly different resonant spectra. The Fano-like shape observed in the reflection spectrum results from interference between the localized high-Q eigenmode and a weakly radiative background channel, a common feature in open resonator systems. For the radial probe aligned with the polarization state, a clear resonance occurs at 1.4494 GHz with a high-quality factor (Q factor) of 5130, which falls within the gain linewidth of the pentacene molecule (see Supplementary Fig. S3). We then experimentally mapped the electric field intensity and polarization angle by positioning the two orthogonal probes at various points, using polar coordinates on a plane parallel to the cube's surface. The resulting maps in Fig. 2e, f confirm experimentally the vortex-like intensity profile and polarization angle winding of the radiated electric field, in excellent agreement with simulation predictions.

### Topological multi-directional vortex masing

To achieve masing action from the cubic cavity, we optically pumped the gain medium using a multimode fiber, which delivers high-power pulses from a laser source to efficiently excite the gain medium within the cube (see "Methods" and Supplementary Fig. S5 for the detailed optical pumping setup). Figure 3a presents the experimentally measured masing spectrum in the frequency domain, demonstrating successful generation of microwave photons at 1.4493 GHz, characterized by a narrow linewidth of 0.165 MHz. The small frequency difference between the reflection and maser spectra arises from the shift in cavity resonance caused by the insertion of the pentacene:p-terphenyl crystals and fine-tuning with metallic mirrors for optimal overlap with the gain medium linewidth (see Supplementary Fig. S2). This narrow linewidth is attributed to the high-Q resonance of the cubic cavity. The masing threshold was determined by varying the pump laser energy and measuring the corresponding output power of masing signals. As shown in Fig. 3b, a pronounced threshold behavior is observed, with an abrupt increase in masing output after the low masing threshold of 4.5 mJ.

On the other hand, the time-domain measurement of the pulses is shown in Fig. 3c. The intensity of the masing signal displays a pulsed envelope that rises and decays within approximately 40 μs. Once the masing signal becomes prominent—typically within a 10–30 μs time window—the associated polarization angle reaches a relatively stable state, as evidenced by the flat plateau highlighted in the shaded region of Fig. 3d, which also implies the presence of stable spatial vortex structures thanks to the coherence of the emission. Under repeated excitation with successive laser pulses, the short-time Fourier transform shown in Fig. 3e captures multiple masing events as hotspots, confirming the stability and robustness of vortex masing under long-term operation, both in the time and frequency domains.

To verify the topological multi-directional vortex masing, Fig. 3f experimentally demonstrates the spatial distribution of the microwave emission in three directions perpendicular to the surfaces of the cubic cavity. This is achieved by mapping the emitted electric field intensity and polarization angle across three distinct cube faces: Face 1 (top), Face 2 (front), and Face 3 (left). The presence of consistent ring-shaped intensity profiles and accompanied polarization winding patterns on all three surfaces confirms the successful realization of coherent multi-directional vortex masing.

We conducted additional experiments to probe the masing signals at different distances and their associated topological polarization singularities. The results are in Supplementary Fig. S10. The observed coexistence of these features confirms our multi-directional vortex maser as a volumetric masing source carrying multidimensional topological singularities. Notably, this topological multi-directional vortex maser represents a significant advancement in higher-dimensional light sources[36], distinguished by its nontrivial spatial structure extending beyond a planar plane to 3D distributions, pulsed

operation, and multi-directional emission. To advance the functionality and reconfigurability of the cubic multi-directional vortex maser, we demonstrate a system comprising two stacked-and-twisted cubic cavities capable of manipulating the masing vortex state on the higher-order Poincaré sphere (see Supplementary Fig. S11), where each point on the sphere represents a distinct topological vortex distribution rather than a single polarization state. Experimentally, we achieved controlled vortex twisting up to 120°, establishing the capability to program multidimensional vortex emissions with tailored topological characteristics. This method provides enhanced control over the spatial features and structured fields in coherent microwave radiation.

### Maser pulses with orbital angular momentum

Having demonstrated maser pulses with polarization vortices, we now construct masing pulses with orbital angular momentum, which necessitates breaking the locally planar phase fronts of the emission. If we picture the linear polarization emitted from our maser as a superposition of right-handed circularly polarized (RCP) and left-handed circularly polarized (LCP) light, we know, due to spin-orbit locking[40,41], that each of these circular components taken separately exhibits an inherently nontrivial OAM with opposite topological charges. When observing the phase under one of the two circular eigenstates, the topological 2π phase manifests itself as an OAM phase vortex (see "Methods" for a theoretical proof). A polarization-selective metasurface can therefore be used to force the maser to emit only one circular polarization, which results in OAM masing. Therefore, we built a chiral metasurface and placed it near our masing cavity to turn the system into a coherent OAM microwave source.

The simulated phase fronts emitting from the cubic cavity exhibit spiral distributions under the two circularly polarized states, as shown in Fig. 4a and d for RCP and LCP, respectively (see also Supplementary Fig. S6 for the topological OAM phase distribution at different positions away from the maser cube). The chiral metasurfaces required to force the maser to emit selected circular polarization states are illustrated in Fig. 4b and e (see "Methods" for details on the design and fabrication of the chiral metasurfaces). Under the same pumping conditions, the phase profile was experimentally reconstructed by performing cross-correlation analysis between time-domain signals from two probes—one scanned across the metasurface while the other remaining fixed as the reference. The experimentally measured phase of the transmitted component exhibits opposite nontrivial topological charges ±1 for RCP and LCP states (Fig. 4c and f), providing direct evidence of the realized topological OAM masing. Note that despite the visual resemblance with previous figures, these panels represent the winding of the phase front, very different from the polarization windings previously reported in absence of metasurface. To access higher-order OAM states, additional engineering strategies are required, such as mode hybridization, where multiple cavity modes with distinct polarization singularities are coupled to form composite eigenstates with larger winding numbers.

## Discussion

In conclusion, we report a multi-directional vortex maser employing a compact, subwavelength cubic cavity that generates volumetric vortex beams through polarization singularities, establishing a new coherent microwave platform for structured wave applications[42]. Our experimental results demonstrate vortex emission from each of the cube surfaces with pulsed operation, high stability, and low-threshold characteristics. Through spin degeneracy lifting induced by a chiral metasurface, the maser produces pulses that carry orbital angular momentum, confirmed by direct measurement of nontrivial phase winding around the beam axis.

This demonstration of a standalone topological vortex emitter enables full exploitation of 3D spatial wave engineering and volumetric wave-matter interactions for communications, sensing, security, and deep-space navigation. The compact design offers advantages for

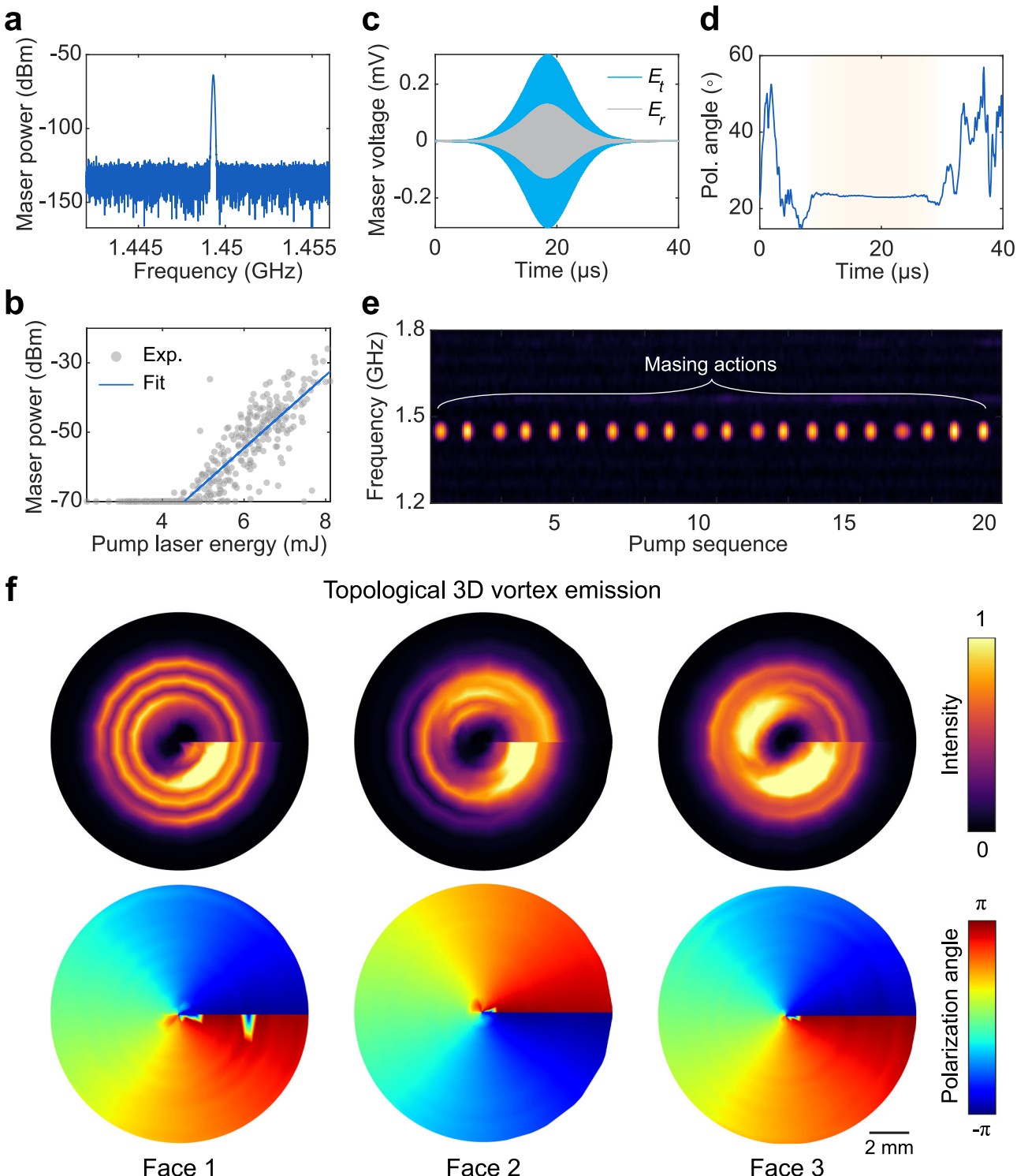

**Fig. 3 | Dynamics of pulsed maser operation. a** Measured maser emission spectrum with microwave photon generation at 1.4493 GHz with a linewidth of 0.165 MHz. **b** Maser output power as a function of pump laser energy. **c** Typical time-domain maser pulses detected by two orthogonal electric probes. **d** The temporal evolution of polarization angle for the emitted maser pulse. **e** Short-time Fourier transform of consecutive maser pulses excited at a pump repetition rate of 5 Hz, revealing the stability and robustness of the pulsed vortex maser. **f** Measured electric field intensity map and polarization angle distribution emitted in three orthogonal directions. Faces 1–3 correspond to top, front, left parallel planes of the cube.

integration and potential as a multi-directional OAM photon source, particularly in quantum optics and entangled vortices[43]. Multi-dimensional topological properties are expected to provide access to disorder-resistant platforms for studying active higher-order topological phenomena[26,44]. Moreover, incorporating improved reconfigurable mechanisms could enable ultrafast manipulation and modulation of vortex charge, on/off switching, and distortion, creating real-time on-demand tunable volumetric vortex generators. Other high-symmetry dielectric resonators, including Platonic solids such as the dodecahedron, can be engineered to support eigenmodes in which the polarization field exhibits singularities aligned with multiple face normals, thereby enabling multi-directional vortex emission.

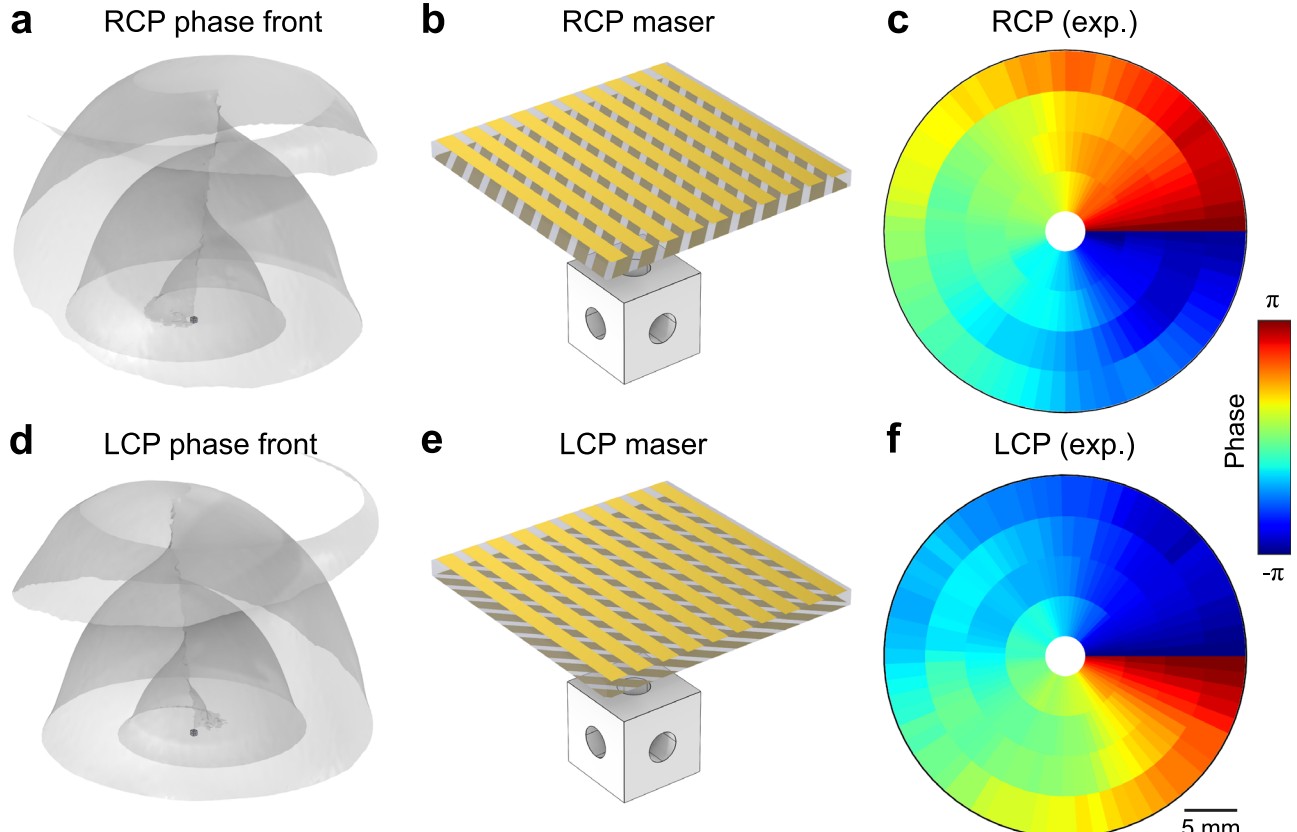

**a** RCP phase front   **b** RCP maser   **c** RCP (exp.)

**d** LCP phase front   **e** LCP maser   **f** LCP (exp.)

5 mm

**Fig. 4 | Observation of maser pulses with orbital angular momentum.**
**a**, **d** Simulated phase front of RCP (**a**) and LCP (**d**) maser modes. **b**, **e** Configuration for inducing spin-selective maser emission using a metasurface placed in the vicinity of the vortex maser, realizing the RCP maser (**b**) and LCP maser (**e**). **c**, **f** Experimentally measured masing phase of RCP (**c**) and LCP (**f**), evidencing the orbital angular momentum carried by the maser pulses.

This concept also offers insights into optical and quantum regimes, where advanced two-photon polymerization lithography can construct 3D cubes and meta-cubes assembled with quantum dots, perovskites, and quantum wells for topological light emission, thereby opening new avenues in structured light generation.

## Methods

### Polarization and phase vortex
Polarization vortices correspond to spatial singularities in the polarization vector field, where the local polarization angle is defined as $\phi_p(\mathbf{r}) = \arg(E_x + iE_y)$, and exhibits a $2\pi m$ winding around a singularity, with $m$ the polarization topological charge and $\mathbf{r}$ the spatial coordinate. In contrast, phase vortices are associated with the azimuthal phase winding of the scalar field amplitude, $\phi(\mathbf{r}) = m\theta$, where $\theta$ is the azimuthal coordinate and $m$ denotes the OAM charge. While polarization vortices are vectorial in nature and stem intrinsically from the cavity eigenmode, phase vortices emerge when spin degeneracy is lifted by the chiral metasurface, which enforces emission in a single circular polarization channel and thereby converts vectorial singularities into OAM states.

According to the Poincaré−Hopf theorem, the sum of topological charges of all vector field singularities on a closed surface must equal the Euler characteristic of the surface. For a topologically equivalent sphere enclosing the cavity, the Euler characteristic is 2. Thus, the total polarization vorticity over any closed surface surrounding the resonator is constrained to be 2, consistent with Fig. 2b.

### Design and fabrication of the STO cube
The cubic cavity was designed using eigenmode solvers in COMSOL Multiphysics for obtaining the optimal magnetic Purcell factor, and vortex electric field. A vortex pattern is confirmed with a continuous transformation of polarization angle from −π to π, corresponding to a topological charge of 1[45]. The evaluation of emissions from the 3D topological eigenmode is simulated through dipole excitation within the cavity. The optimized cavity has a length of 14.1 mm and a hole radius of 2.5 mm, which was then fabricated in bulk SrTiO$_3$ with all surfaces polished by Hefei Kejing Materials Technology Co., Ltd.

### Pentacene: p-terphenyl crystal growth
The pentacene-doped p-terphenyl crystals (0.1% mol/mol) are grown using the vertical Bridgman method. Initially, commercially purchased p-terphenyl (A14833, Alfa Aesar) is purified by sublimation under high-vacuum ($\leq 10^{-6}$ mbar) sublimation purification and then mixed with as-received pentacene powder (P2524, TCI Europe NV). The mixture is loaded into a double-layered ampoule (5 mm inner diameter), and a 1 mm diameter Pyrex rod is inserted to create a central hole through the crystal. The ampoule is then sealed under high vacuum and subjected to a controlled growth process in a Bridgman furnace: 12 h of melting at 240 °C for homogenization, followed by a 14-day descent at 0.53 mm/h towards the 80 °C zone. After growth, the furnace is gradually cooled down to room temperature. Finally, the crystal is cut using a diamond saw and separated from the ampoule and rod (see Supplementary Fig. S2).

### Design and fabrication of metasurfaces
The polarization singularities introduce vectorial vortices that possess inherent spin-orbit locking, which can be decomposed by two circular states as $|E_{(\phi)}\rangle = e^{i\phi}|R\rangle + e^{-i\phi}|L\rangle$, where $|R\rangle$, $|L\rangle$ are the RCP and LCP states and $\phi$ is the polarization angle. A nontrivial winding of $\phi$ results in opposite topological charges in phase components of RCP and LCP[40,41]. This lays the foundation for showing OAM pulses from the vortex

maser. Simulation integrating STO cube and metasurface for blocking RCP or LCP component is conducted and verified in COMSOL Multiphysics (see Supplementary Fig. S7). Efficient OAM phase is detected after vector vortex passes through the metasurface. The metasurfaces are fabricated on a low-loss dielectric substrate (Taizhou Wangling, F4BTMS1000) with permittivity of 10.2 and tangential loss of 0.002. Copper gratings are deposited on double sides of each substrate. The fabricated metasurfaces have a thickness of 6.35 mm and side length of 108 mm (see Supplementary Figs. S8 and S9).

## Optical setups

A schematic optical path is illustrated in Supplementary Fig. S5. The output beam from an optical parametric oscillator is divided using a beam splitter (15%:85%): the transmitted beam is shrunk using a telescope and measured by a power meter (Gentec-EO, Pyroelectric detector); the reflected beam is coupled into a multi-mode optical fiber (core diameter of 1 mm) using a plano-convex lens. To filter the spatial profile of the focused beam and protect the optical fiber, a precise pinhole is placed in front of the optical fiber. The other end of the optical fiber enters the pentacene:p-terphenyl (pc:pt) crystal for efficient optical pumping.

## Maser characterization

Passive and active characterization of the maser platform are conducted using a vector network analyzer (Rohde & Schwarz ZNA67) with a magnetic probe for resonant frequency tuning and an oscilloscope (Tektronix) with electric probes for measurement of time-domain masing pulses, respectively.

## Masing field mapping

Two orthogonal electric field probes, mounted on a motorized rotary arm with an adjustable radius, were utilized to map the vortex-masing field distribution in polar coordinates. The amplitude and polarization of the electric field at each position were determined by analyzing the ratio of masing amplitudes and phase difference between the two probes. To retrieve the masing phase vortex, one probe is fixed, and the other is sweeping across a surface above the metasurface. The relative phase between two probed fields can be extracted from the cross-correlation operation between time-domain pulsed signals and finding the maximum lags.

## Data availability

The data supporting the findings of this study are presented within the article and Supplementary Information.

## Code availability

The code used to evaluate the conclusions in the paper is available from the corresponding authors upon request.

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

## Acknowledgements

This work was supported by the Swiss National Science Foundation under the Eccellenza grant 181232 and the Swiss State Secretariat for Education, Research and Innovation (SERI) under contract No. MB22.00028.

## Author contributions

R.F., H.Q., and R.X. conceived the project. R.X. and H.Q. carried out numerical simulation, fabricated the sample, and conducted the measurements. A.J. designed the metasurfaces. H.Q., R.X., Z.Z., and R.F. wrote the manuscript. R.F. supervised the entire project. All authors discussed the results and contributed to the manuscript.

## Competing interests

The authors declare no competing interests.
