## [Transparent Peer Review file · Nature Communications]

Topological radiation from vortex masers

Corresponding Author: Professor Romain Fleury

Version 0:

Reviewer comments:

Reviewer #1

(Remarks to the Author)

This manuscript presents a study on topological radiation from vortex masers, demonstrating a room-temperature pulsed maser with polarization vortices based on a subwavelength cubic strontium titanate (STO) cavity. Specifically, a chiral metasurface is used to decouple circular polarizations, thereby realizing the masing of phase vortices. I think the topic is interesting and the paper is well-written. Both theoretical and experimental results are well presented. I have some minor questions/comments that need to be clarified:

1. The authors mentioned in the ABSTRACT that the room-temperature maser emitting pulses of electromagnetic radiation with polarization and phase vortices. It is recommended that the authors supplement detailed elaboration on the polarization vortices and phase vortices, as well as their differences.
2. In the manuscript, the eigenmode of cubic STO cavity exhibit a zero-intensity singularity at the center and a 2π polarization angle winding. Could the authors elaborate on two points: First, why the 2π polarization angle winding corresponds to a topological charge of ± 1 ; Second, why the total instantaneous polarization vorticity over any surface enclosing the resonator is 2?
3. The cubic STO cavity generates vortices on all faces, relying on its high structural symmetry. Can a dodecahedral STO cavity achieve vortex masing with more emission directions?
4. The decoupling effect of the chiral metasurface is key to generating OAM. However, the current version of the manuscript only mentions the metasurface's material and its functional role, without providing any simulation results related to the metasurface itself.
5. Is there a way to generate OAM without metasurface? When a metasurface is adopted, the STO cavity lacks advantages—this is because the metasurface itself is capable of generating OAM directly .
- 6, How to tune the emission frequencies of the maser?
- 7, Can the maser generate OAM with arbitrary topological charges?

Reviewer #2

(Remarks to the Author)

In this manuscript, the authors have carefully studied the vortex maser emission from a cubic SrTiO₃ cavity with pentacene crystals. Under an optical excitation at 592 nm, vortex laser emission has been achieved in all three dimensions. The entire cavity is much smaller than the wavelength. I have the following comments for the authors.

1. The fundamental physics of this research is the 3D topological vectorial singularities. This point has been shown with numerical simulation in both main text and Fig. S1. But the corresponding mechanism has not been discussed yet. A physical picture is suggested to highlight the novelty of this research.
2. The size information of the cavity is missing. It should be added to either the figure caption or in the main text. Scale bar

should also be added to all the images.

3. The resonant position in reflection spectrum and maser spectrum is slightly different (~ 0.1 MHz, close to the laser linewidth). What is the reason? Meanwhile, the reflection spectrum shows a fano shape. This has not been discussed yet.

4. In page 5, the authors stated "they create a special resonance with multiple topological polarization singularities in all normal directions". This sentence is quite confusing. I think the authors try to discuss the singularity in all three directions. The corresponding topological charge should be added to avoid confusion.

5. For the case of transition from vortex laser to vortex maser, the absence of vortex maser cannot show the novelty and importance of this research. Maser operates at a much longer wavelength than laser. It is well known that almost all the techniques in lasers can be applied to maser too. Some techniques that are missing in laser can also be realized in maser, e.g., three dimensional control in this research. The authors should provide a more convincing reason for this research.

Version 1:

Reviewer comments:

Reviewer #1

(Remarks to the Author)

The authors have addressed all my comments and revised the manuscript accordingly, I am happy to recommend it for publication in Nature Communications.

Reviewer #2

(Remarks to the Author)

I have carefully read the response letter and revised manuscript. The authors have clearly answered my questions. Even though I am not satisfied with the reply about laser and maser comment, I think the scientific finding in this research can be considered for publication now.

Reply to Reviewer #1

Comment: This manuscript presents a study on topological radiation from vortex masers, demonstrating a room-temperature pulsed maser with polarization vortices based on a subwavelength cubic strontium titanate (STO) cavity. Specifically, a chiral metasurface is used to decouple circular polarizations, thereby realizing the masing of phase vortices. I think the topic is interesting and the paper is well-written. Both theoretical and experimental results are well presented. I have some minor questions/comments that need to be clarified:

Response: We sincerely thank you for the positive assessment of our work and for highlighting the interest and clarity of our manuscript. Below, we provide detailed clarifications in response to your minor comments.

Comment 1: The authors mentioned in the ABSTRACT that the room-temperature maser emitting pulses of electromagnetic radiation with polarization and phase vortices. It is recommended that the authors supplement detailed elaboration on the polarization vortices and phase vortices, as well as their differences.

Response: We thank you for this valuable suggestion.

In the revised manuscript, we have clarified the distinction between polarization vortices and phase vortices with explicit mathematical descriptions. Polarization vortices correspond to spatial singularities in the polarization vector field, where the local polarization angle is defined as

$$\phi_p(\mathbf{r}) = \arg(E_x + iE_y),$$

and exhibits a $2\pi m$ winding around a singularity, with m the polarization topological charge. In contrast, phase vortices are associated with the azimuthal phase winding of the scalar field amplitude,

$$\phi(\mathbf{r}) = m\theta,$$

where θ is the azimuthal coordinate and m denotes the orbital angular momentum (OAM) charge. While polarization vortices are vectorial in nature and stem intrinsically from the cavity eigenmode, phase vortices emerge when spin degeneracy is lifted by the chiral metasurface, which enforces emission in a single circular polarization channel and thereby converts vectorial singularities into OAM states.

These distinctions have been explicitly added to the revised manuscript as a new section in **Methods: Polarization and phase vortex.**

Comment 2: In the manuscript, the eigenmode of cubic STO cavity exhibit a zero-intensity singularity at the center and a 2π polarization angle winding. Could the authors elaborate on two points: First, why the 2π polarization angle winding corresponds to a topological charge of ± 1 ; Second, why the total instantaneous polarization vorticity over any surface enclosing the resonator is 2?

Response: We thank the reviewer for raising these important points.

We thank you for raising this important question, which allows us to clarify the topological interpretation of our results.

a) Topological charge of ± 1 .

The polarization angle is defined as

$$\phi_p(\mathbf{r}) = \arg(E_x + iE_y),$$

Around a singularity where the field amplitude vanishes, the phase of the polarization vector winds continuously as one traverses a closed contour. The topological charge q is given by the winding number:

$$q = \frac{1}{2\pi} \oint_{\Gamma} d\phi_p,$$

where Γ is a closed loop around the singularity. A single 2π winding corresponds to $q = \pm 1$, with the sign determined by the direction (clockwise or counterclockwise) of the winding. This definition is standard for polarization singularities and matches our simulation and experimental observations.

b) Total polarization vorticity of 2.

The cubic eigenmode hosts multiple polarization singularities embedded in three-dimensional space. According to the Poincaré–Hopf theorem, the sum of topological charges of all vector field singularities on a closed surface must equal the Euler characteristic of the surface. For a topologically equivalent sphere enclosing the cavity, the *Euler characteristic* is 2. Thus, the total polarization vorticity over any closed surface surrounding the resonator is constrained to be

$$\sum_i q_i = 2,$$

consistent with our calculated eigenmode, which exhibits two unit-charge singularities when summing over each cut-plane, $1+1+(-1) = 1$ for Fig. 2b and the other three surfaces to 1, finally leading to an overall charge of 2. This conservation ensures that the observed singularities are robust and topologically protected.

We have added these explanations to the revised manuscript as a new section in **Methods: Polarization and phase vortex**.

Comment 3: The cubic STO cavity generates vortices on all faces, relying on its high structural symmetry. Can a dodecahedral STO cavity achieve vortex masing with more emission directions?

Response: We really appreciate your forward-looking question. In principle, yes—other high-symmetry dielectric resonators (including Platonic solids such as a dodecahedron) can be engineered to support eigenmodes whose polarization field exhibits singularities aligned with multiple face normals, enabling multi-directional vortex emission. The key requirement is not the cube per se but the existence of a 3D eigenmode with (i) ring-like intensity and a central zero on planes parallel to the emitting faces, and (ii) a nontrivial winding of the local polarization angle around those zeros. One concern may be that the dodecahedral boundary will reduce the Q factor if surface curvature or machining tolerances increase radiation or ohmic loss.

In short, a dodecahedral STO cavity is a promising extension and could yield a richer set of emission directions if one designs an eigenmode whose polarization singularities are oriented along the desired face normal. We have added a brief discussion of this potential in the outlook (Page 12) of the revised manuscript:

“Other high-symmetry dielectric resonators, including Platonic solids such as the dodecahedron, can be engineered to support eigenmodes in which the polarization field exhibits singularities aligned with multiple face normals, thereby enabling multi-directional vortex emission.”

Comment 4: The decoupling effect of the chiral metasurface is key to generating OAM. However, the current version of the manuscript only mentions the metasurface’s material and its functional role, without providing any simulation results related to the metasurface itself.

Response: We thank you for pointing out this important aspect. In the revised manuscript, we now provide additional simulation results and explanations regarding the metasurface’s decoupling function. Specifically, the metasurface enforces spin-selective transmission by blocking either the RCP or LCP component of the incident field. This functionality was validated using full-wave simulations.

The simulated spiral phase fronts are shown in Supplementary Fig. S7, where the metasurface selectively converts vectorial polarization vortices into scalar OAM vortices with opposite topological charges for RCP and LCP channels.

These additional results and explanations have been incorporated into the **Methods: Design and fabrication of metasurfaces** and **Supplementary Information (Figs. S6-S9)**.

Comment 5: Is there a way to generate OAM without metasurface? When a metasurface is adopted, the STO cavity lacks advantages—this is because the metasurface itself is capable of generating OAM directly.

Response: We appreciate your very thoughtful question.

a) Regarding OAM without a metasurface.

In principle, one may achieve intrinsic spin selection in the resonator by (i) introducing weak geometric chirality (boundary/perturbations), (ii) adding gyrotropy (e.g., magneto-optic materials), or (iii) engineering non-Hermitian coupling to favor a single circular eigenchannel. We emphasize, however, that such approaches typically increase system complexity, reduce the quality factor, raise the masing threshold, and may challenge or hinder uniform optical pumping as well as Purcell optimization.

b) The concern that “the metasurface itself can generate OAM directly”.

Metasurfaces designed as helical phase plates or PB holograms can indeed create OAM from a trivial input, but that is not the component we use here. Our device uses a compact spin-selective grating whose role is analyzable selection (RCP/LCP) rather than spatially varying phase imprinting. The central focus of this work is the realization of a nontrivial, multidimensional *coherent microwave source*.

We have added this point in Page 10

“A polarization-selective metasurface can therefore be used to force the maser to emit only one circular polarization, which results in OAM masing. Therefore, we built a chiral metasurface and placed it near our masing cavity to turn the system into a coherent OAM microwave source.”

Comment 6: How to tune the emission frequencies of the maser?

Response: We thank you for this pertinent question. In our system, the maser emission frequency is primarily determined by the cavity resonance of the subwavelength STO cube, which must overlap with the masing transition between the spin sublevels of the pentacene: p-terphenyl (pc:pt) gain medium. Tuning the emission frequency requires reconfiguring the cavity, providing one remains within the gain bandwidth. Larger reconfigurability requires tuning the cavity frequency and modifying the sublevels, which can be achieved by imparting a small magnetic field to the gain medium.

We can explicit several strategies to fine-tune the resonance frequency:

Geometric tuning. Varying the cube dimension or hole radius shifts the eigenmode frequency, offering fine geometric control during fabrication.

Dielectric tuning. Since the cavity resonance scales with the permittivity, temperature control of the high-index STO host material provides an efficient tuning mechanism. Small temperature variations induce measurable frequency shifts due to the strong dielectric dispersion of STO in the microwave regime.

Magnetic or strain perturbations. Weak symmetry-breaking perturbations—such as inserting dielectric rods, applying strain, or local loading with magneto-optic materials—can lift mode degeneracies and shift resonance frequencies, while still preserving the topological vortex characteristics.

In the Supplementary Fig. S2, we also show the capability of frequency tuning via moveable metallic plates around the cubic cavity, acting as a perturbation:

“Configuration of the vortex maser experiment setup with two-sidewall metal holder and two movable plates for fine-tuning resonance of cubic cavity, enabling masing vortices in three dimensions and related measurement.”

Comment 7: Can the maser generate OAM with arbitrary topological charges?

Response: We thank you for this insightful question. In the current design, the maser generates OAM states with topological charge of ± 1 , which directly arises from the spin-orbit-locked decomposition of the cavity’s vector vortex eigenmode into its RCP and LCP components.

To access higher-order OAM states, additional engineering is required. Possible strategies include *mode hybridization*, in which multiple cavity modes with distinct polarization singularities are coupled to form composite eigenstates with larger winding numbers, and the use of *stacked or twisted cubes*. As demonstrated in Supplementary Fig. S11, two stacked and rotated STO cubes enable controlled twisting of the vortex emission on the higher-order Poincaré sphere, a concept that may be further extended to synthesize higher-order OAM distributions.

We have clarified in the revised manuscript that while the present device demonstrates coherent OAM masing with charge 1, there is no fundamental restriction of the underlying concept with respect to extensions toward arbitrary higher-order charges through cavity engineering.

We added this point in Page 1.

“To access higher-order OAM states, additional engineering strategies are required, such as mode hybridization, where multiple cavity modes with distinct polarization singularities are coupled to form composite eigenstates with larger winding numbers.”

Reply to Reviewer #2

Comment: In this manuscript, the authors have carefully studied the vortex maser emission from a cubic SrTiO₃ cavity with pentacene crystals. Under an optical excitation at 592 nm, vortex laser emission has been achieved in all three dimensions. The entire cavity is much smaller than the wavelength. I have the following comments for the authors.

Response: We sincerely thank you for the constructive suggestions and comments. We have provided point-by-point responses below and carefully revised the manuscript to include the requested clarifications and additional details where appropriate.

Comment 1: The fundamental physics of this research is the 3D topological vectorial singularities. This point has been shown with numerical simulation in both main text and Fig. S1. But the corresponding mechanism has not been discussed yet. A physical picture is suggested to highlight the novelty of this research.

Response: We thank you for this insightful comment. In the revised manuscript, we have added a clear physical picture to highlight the mechanism and novelty of the 3D topological vectorial singularities. The cubic STO cavity supports a unique eigenmode in which the electric field intensity vanishes at the center ($E_x = 0$ and $E_y = 0$ forming nodal lines and crossing at the center point as vectorial zero), while the polarization angle $\phi_p(\mathbf{r}) = \arg(E_x + iE_y)$ undergoes a continuous 2π winding on planes parallel to the cube faces. This configuration gives rise to polarization vortices with unit topological charge in multiple directions. Importantly, the distribution of these singularities is not accidental—it is mandated by the Poincaré–Hopf theorem, which enforces that the total polarization index over a closed surface enclosing the resonator must equal 2. This ensures that the observed vortices are robust, symmetry-protected, and cannot be removed by small perturbations.

Physically, this eigenmode can be understood as a topologically twisted vectorial resonance: a dipolar-like mode reshaped by cubic symmetry into a three-dimensional configuration where polarization singularities are simultaneously radiated toward all six faces. The drilled voids concentrate magnetic fields, ensuring efficient coupling with the spin-polarized triplet transitions in the pentacene:p-terphenyl gain medium. This synergy between cavity symmetry, vectorial topology, and gain–cavity interaction constitutes the physical foundation of the maser’s multidirectional vortex emission.

We believe that this physical interpretation not only clarifies the origin of the 3D vectorial singularities but also underscores the novelty of our work: the first realization of a coherent volumetric source where topological polarization singularities are intrinsically embedded in the eigenmode itself, rather than imposed externally. This discussion has been incorporated into the revised manuscript in Page 5,

“The cubic STO cavity supports a unique eigenmode in which the electric field intensity vanishes at the center ($E_x = 0$ and $E_y = 0$ forming nodal lines and crossing at the center point as vectorial zero), while the polarization angle $\phi_p(\mathbf{r}) = \arg(E_x + iE_y)$ undergoes a continuous 2π winding on planes parallel to the cube faces. The drilled voids concentrate magnetic fields, ensuring efficient coupling with the spin-polarized triplet transitions in the gain medium. This synergy between cavity symmetry, vectorial topology, and gain–cavity interaction constitutes the physical foundation of the maser’s multidirectional vortex emission. This provides a physical picture of the mechanism underlying the 3D topological singularities, underscoring the novelty of our work: the first realization of a coherent volumetric source where topological polarization singularities are intrinsically embedded in the eigenmode itself, rather than imposed externally.”

Comment 2: The size information on the cavity is missing. It should be added to either the figure caption or in the main text. Scale bar should also be added to all the images.

Response: We thank you for pointing this out. We have included the precise size information of the cubic STO cavity in both the Methods section and the relevant figure caption in Fig. 2.

“The cavity has a side length of 14.1 mm and a hole radius of 2.5 mm.”

In addition, scale bars have now been added to all optical and simulated field distribution images to provide clear dimensional reference.

We believe these additions improve the clarity and completeness of the manuscript.

Comment 3: The resonant position in reflection spectrum and maser spectrum is slightly different (~ 0.1 MHz, close to the laser linewidth). What is the reason? Meanwhile, the reflection spectrum shows a fano shape. This has not been discussed yet.

Response: We thank you for these questions. The small frequency difference between the reflection resonance and the maser emission arises from the measurement conditions. The reflection spectrum was taken with the hollow cubic cavity prior to inserting the gain medium. Once the pentacene:p-terphenyl crystals are inserted, their dielectric index slightly shifts the cavity resonance, accounting for the ~ 0.1 MHz difference. In addition,

during active operation we employed metallic mirrors to fine-tune the resonance frequency, ensuring optimal overlap with the gain medium linewidth. These combined effects explain the observed shift.

Regarding the Fano-like line shape observed in the reflection spectrum, the resonance couples simultaneously to a localized high-Q eigenmode and a weakly radiative background channel, producing the characteristic asymmetric Fano profile. This interference effect is expected in open resonator systems.

We have clarified these two points in the revised manuscript:

In Page 8, “The small frequency difference between the reflection and maser spectra arises from the shift in cavity resonance caused by the insertion of the pentacene:p-terphenyl crystals and fine-tuning with metallic mirrors for optimal overlap with the gain medium linewidth (see Fig. S2).”

In Page 7, “The Fano-like shape observed in the reflection spectrum results from interference between the localized high-Q eigenmode and a weakly radiative background channel, a common feature in open resonator systems.”

Comment 4: In page 5, the authors stated "they create a special resonance with multiple topological polarization singularities in all normal directions". This sentence is quite confusing. I think the authors try to discuss the singularity in all three directions. The corresponding topological charge should be added to avoid confusion.

Response: We thank you for pointing out this ambiguity. In the revised manuscript, we have clarified the description of the eigenmode singularities. Specifically, the drilled voids in the cubic cavity create a resonance that hosts polarization singularities along all three orthogonal directions (normal to the cube faces). On each cut-plane, the polarization angle $\phi_p(\mathbf{r}) = \arg(E_x + iE_y)$ exhibits a continuous 2π winding around the zero-intensity center, corresponding to a topological charge of 1. These singularities are replicated on the three mutually orthogonal planes, ensuring multi-directional vortex emission.

We have revised the sentence in page 5, paragraph 2 to explicitly include the topological charge and to state that the singularities occur along the three orthogonal directions, thereby eliminating the confusion:

In Page 6, “It’s noteworthy that the functionality of the air holes is twofold: they create a special resonance that hosts polarization singularities along all three orthogonal directions (normal to the cube faces), while simultaneously providing a void space with concentrated magnetic field, maximizing the interactions with the magnetic dipoles in the gain medium.

When viewed at any cut plane parallel to the cube's surface, the simulated electric field profile consistently exhibits a ring-shaped distribution with a zero singularity at the center and high amplitude in the surrounding region (Fig. 2a). The resultant polarization angle map reveals a 2π winding around each singularity (corresponding to a topological charge of ± 1), confirming the vortex nature of the electric eigenmode in multiple directions (Fig. 2b).”

Comment 5: For the case of transition from vortex laser to vortex maser, the absence of vortex maser cannot show the novelty and importance of this research. Maser operates at a much longer wavelength than laser. It is well known that almost all the techniques in lasers can be applied to maser too. Some techniques that are missing in laser can also be realized in maser, e.g., three dimensional control in this research. The authors should provide a more convincing reason for this research.

Response: We thank you for raising this important point regarding the motivation and significance of our work.

The realization of a 3D vortex maser is a significant advancement because it extends the principles of **active topological photonics**—previously limited to optical lasers—into the microwave regime, where coherent emission has not been achieved at this level of complexity. Unlike lasers, which typically operate with vortex emission confined to 2D geometries (e.g., planar metasurfaces), microwaves operate at much longer wavelengths, making it more challenging to achieve subwavelength cavities capable of supporting complex, multidimensional topological structures.

Our work demonstrates the **first room-temperature maser** that produces **3D polarization vortices** within a **subwavelength cubic STO cavity**, achieving **multidirectional vortex emission** that is also not possible in conventional laser systems.

The **coherent emission of these vortex states** in the microwave range is crucial because, unlike optical light, microwaves at room temperature can interact with materials and structures in ways that allow for **highly sensitive measurements, microwave quantum devices, and advanced communication systems**. The ability to generate coherent, topologically structured microwave radiation opens up new avenues for **quantum computing, microwave imaging, and signal processing** where such coherent states are essential for enhancing performance, reducing noise, and enabling more precise control over electromagnetic fields.

The topological singularities in our system are intrinsic to the eigenmode itself, providing enhanced flexibility and stability compared to systems where the topology is externally imposed. This marks a key advancement in maser technology, enabling the generation of complex microwave light fields that are critical for applications requiring fine control over light-matter interactions in the microwave regime.

By demonstrating 3D vortex emission in a compact, room-temperature maser, we not only extend topological wave engineering to the microwave domain but also create a new platform for future nanophotonic, quantum, and communication devices.

We have expanded the introduction and conclusion of the revised manuscript to emphasize these aspects and to make the novelty and importance of this research clearer.

In Page 4, “This extends the principles of active topological photonics, previously confined to optical lasers, into the microwave regime, where coherent emission of this complexity has not been achieved. It provides critical insights into how such coherent states can enhance performance, reduce noise, and enable more precise control over electromagnetic fields.”

In Page 6, “The topological singularities in our system are intrinsic to the eigenmode itself, providing enhanced flexibility and stability compared to systems where the topology is externally imposed.”

In Page 12, “This concept also offers insights into optical and quantum regimes, where advanced two-photon polymerization lithography can construct 3D cubes and meta-cubes assembled with quantum dots, perovskites, and quantum wells for topological light emission, thereby opening new avenues in structured light generation.”

Reviewer #1 (Remarks to the Author):

The authors have addressed all my comments and revised the manuscript accordingly, I am happy to recommend it for publication in Nature Communications.

Our reply: We thank the reviewer for their positive evaluation and recommendation for publication.

Reviewer #2 (Remarks to the Author):

I have carefully read the response letter and revised manuscript. The authors have clearly answered my questions. Even though I am not satisfied with the reply about laser and maser comment, I think the scientific finding in this research can be considered for publication now.

Our reply: We appreciate the reviewer's careful reading of our revision and constructive feedback. We are grateful that the reviewer recognizes the significance of our findings and supports publication.